# SARS-CoV-2 infection induces mixed M1/M2 phenotype in circulating monocytes and alterations in both dendritic cell and monocyte subsets

Sanja Matic[1], Suzana Popovic[2]*, Predrag Djurdjevic[3,4], Danijela Todorovic[5], Natasa Djordjevic[6], Zeljko Mijailovic[7,8], Predrag Sazdanovic[9,10], Dragan Milovanovic[6,11], Dejana Ruzic Zecevic[6,11], Marina Petrovic[3,12], Maja Sazdanovic[13], Nenad Zornic[14,15], Vladimir Vukicevic[15], Ivana Petrovic[16], Snezana Matic[16], Marina Karic Vukicevic[17], Dejan Baskic[2,18]

1 Department of Pharmacy, Faculty of Medical Sciences, University of Kragujevac, Kragujevac, Serbia, 2 Centre for Molecular Medicine and Stem Cell Research, Faculty of Medical Sciences, University of Kragujevac, Kragujevac, Serbia, 3 Department of Internal Medicine, Faculty of Medical Sciences, University of Kragujevac, Kragujevac, Serbia, 4 Clinic for Haematology, Clinical Centre Kragujevac, Kragujevac, Serbia, 5 Department of Genetics, Faculty of Medical Sciences, University of Kragujevac, Kragujevac, Serbia, 6 Department of Pharmacology and Toxicology, Faculty of Medical Sciences, University of Kragujevac, Kragujevac, Serbia, 7 Department of Infectious Diseases, Faculty of Medical Sciences, University of Kragujevac, Kragujevac, Serbia, 8 Department of Infectious Diseases, Clinical Centre Kragujevac, Kragujevac, Serbia, 9 Department of Anatomy, Faculty of Medical Sciences, University of Kragujevac, Kragujevac, Serbia, 10 Gynecology and Obstetrics Clinic, Clinical Centre Kragujevac, Kragujevac, Serbia, 11 Department of Clinical Pharmacology, Clinical Centre Kragujevac, Kragujevac, Serbia, 12 Clinic for Pulmonology, Clinical Centre Kragujevac, Kragujevac, Serbia, 13 Department of Histology and Embryology, Faculty of Medical Sciences, University of Kragujevac, Kragujevac, Serbia, 14 Department of Surgery, Faculty of Medical Sciences, University of Kragujevac, Kragujevac, Serbia, 15 Corona Centre, Clinical Centre Kragujevac, Kragujevac, Serbia, 16 Department of Microbiology, Clinical Centre Kragujevac, Kragujevac, Serbia, 17 Department of Rheumatology, Allergology and Clinical Immunology, Clinical Centre Kragujevac, Kragujevac, Serbia, 18 Public Health Institute, Kragujevac, Kragujevac, Serbia

* suzana.popovic@medf.kg.ac.rs

**Data Availability Statement:** All relevant data are within the paper and its Supporting Information files.

## Abstract

Clinical manifestations of SARS-CoV-2 infection range from mild to critically severe. The aim of the study was to highlight the immunological events associated with the severity of SARS-CoV-2 infection, with an emphasis on cells of innate immunity. Thirty COVID-19 patients with mild/moderate symptoms and 27 patients with severe/critically severe symptoms were recruited from the Clinical Center of Kragujevac during April 2020. Flow cytometric analysis was performed to reveal phenotypic and functional alterations of peripheral blood cells and to correlate them with the severity of the disease. In severe cases, the number of T and B lymphocytes, dendritic cells, NK cells, and HLA-DR-expressing cells was drastically decreased. In the monocyte population proportion between certain subsets was disturbed and cells coexpressing markers of M1 and M2 monocytes were found in intermediate and non-classical subsets. In mild cases decline in lymphocyte number was less pronounced and innate immunity was preserved as indicated by an increased number of myeloid and activated dendritic cells, NK cells that expressed activation marker at the same level as in control and by low expression of M2 marker in monocyte population. In patients

**Funding:** The authors received no specific funding for this work.

**Competing interests:** The authors have declared that no competing interests exists.

with severe disease, both innate and adoptive immunity are devastated, while in patients with mild symptoms decline in lymphocyte number is lesser, and the innate immunity is preserved.

## Introduction

Severe acute respiratory syndrome coronavirus 2 (SARS-CoV-2) continues to infect millions of people worldwide, causing coronavirus disease (COVID-19). The severity of reported symptoms for COVID-19 ranges from mild to critically severe having significant potential for fatal outcome. Previous studies have revealed a certain pattern of changes in biochemical and hematological parameters, while researches on immunopathology underlying COVID-19 are in progress. Currently, there is no wide agreement of the scientific and medical community about diagnostic, treatment and prognostic importance of immunological parameters for routine practice [1–3].

In COVID-19 patients inflammatory factors such as C–reactive protein (CRP) and erythrocyte sedimentation rate (ESR) are generally elevated, and CRP level, in general, positively correlates with the severity of the infection. High procalcitonin (PCT) level is a highly specific marker of the presence of bacterial infection and elevated levels of aspartate aminotransferase (AST), alanine aminotransferase (ALT), lactate dehydrogenase (LDH), creatine kinase (CK), D-dimer and prothrombin time were proposed to be markers of severe disease [4]. The majority of studies conducted so far found that interleukin-6 (IL-6) serum concentrations positively correlate with exacerbation of disease after 7–14 days of onset of symptoms [5]. Other studies indicate that an increased number of neutrophils in combination with lymphopenia and consequent increase in neutrophil-to-lymphocyte ratio was the prognostic factor for severe cases [6]. The results of the first research on lymphocyte populations' change in severe COVID-19 cases indicated a decreased number of T lymphocytes, an increased number of naive T helper cells, and a decrease in memory T helper cells [1]. Also, the number of CD8+T cells, B cells, and natural killer (NK) cells were substantially reduced in COVID-19 patients, particularly in severe cases [5–7]. In COVID-19 patients with severe pulmonary inflammation expression of NKG2 marker on NK cells and cytotoxic T lymphocytes were markedly increased and tend to correlate with functional impairment, indicating disease progression [8]. Although neutrophilia and impairment of lymphocyte number and function in COVID-19 patients have been well described, fewer data are available on dendritic cells and monocytes [9]. Two groups of authors separately pointed to alterations in the activation status and morphology of monocytes in severe cases. They identified forward scatter high (FCS-high) [10] and side scatter high (SSC-high) [11] populations of monocytes that secrete IL-6, IL-10 and TNF-α. One paper described the depletion of plasmacytoid dendritic cells in patients with severe disease [12].

Detailed analysis of immune parameters in COVID-10 patients and a better understanding of features of immune response underlying distinctive courses of the disease will improve diagnostics, the prognosis of disease outcome, as well as treatment strategies. Here, we present novel observations about changes in morphology and activation status of the cells of the innate immunity in COVID-19 patients, which seem to correlate with the severity of the infection.

## Patients and methods

### Patients/study design and participants

Fifty-seven cases of COVID-19 patients who were hospitalized in the Clinical Center of Kragujevac were recruited in this study during April 2020. Inclusion criteria were: adults of male or

female gender (≥18 years old), SARS-CoV-2 infection confirmed by real-time polymerase chain reaction (RT-PCR) and hospitalization. COVID-19 patients were diagnosed according to the World Health Organization's (WHO) interim guidance [13]. Clinical condition severity was classified in four categories: a) mild; mild clinical symptoms of upper respiratory tract viral infection; b) moderate: present signs of pneumonia without need for supplemental oxygen; c) severe: fever or suspected respiratory infection with compromised respiratory function; and d) critically severe: worsening of respiratory symptoms with the necessity for mechanical ventilation. Disease severity was assessed upon discharge from the hospital. Our cohort of 57 COVID-19 patients consisted of 30 cases of mild/moderate disease (MD) and 27 cases of severe/critical disease (SD). Five healthy subjects with a negative RT-PCR test for coronavirus were included in the study as a control group. Ethics Committee of Clinical Center Kragujevac approved this study (Nr 01/20-405) and prior initiation written informed consent was obtained from every subject or the patients' legal representative if he or she was unable to communicate e.g. sedated on mechanical ventilation, according to the Declaration of Helsinki of the World Medical Association.

## Data collection

The patients' data were obtained from hospital medical records (electronic and paper version) for each study subject according to the modified case record rapid recommendation concerning the patients with Covid-19 infection of WHO (Global COVID-19 Clinical Platform: novel coronavirus (Covid-19)—rapid version. Geneva: World Health Organization, 2020. (https://apps.who.int/iris/rest/bitstreams/1274888/retrieve). The data for the following variables were collected: age, sex, medical history, symptoms, and signs of Covid-19, severity assessment, radiological imaging, and laboratory findings. Blood samples for laboratory tests and flow cytometry analysis were collected on the first day of hospitalisation, before any treatment. The following hematological parameters were examined: white blood cell count (WBC) with WBC differential count (neutrophils, eosinophils, basophils, lymphocytes, and monocytes), red cell blood count (RBC), hemoglobin concentration and platelet count (PLT). The measured biochemical parameters included creatinine (CRE), blood urea nitrogen [3], glycemia (GLY), albumin (ALB), aspartate aminotransferase (AST), alanine aminotransferase (ALT), lactate dehydrogenase (LDH), creatine kinase (CK), D-dimer, C-reactive protein (CRP) and procalcitonin (PCT). Analysis of hematological parameters and blood biochemical assays were performed with commercial reagents and according to good laboratory practices at the hematology and clinical biochemistry departments of the hospital.

## Flow cytometry

EDTA whole blood samples from 5 healthy subjects, 10 patients with mild and 10 patients with severe disease were stained with anti-HLA-DR PE and ECD, anti-CD3 ECD, anti-CD4 PE, anti-CD8 FITC, anti-CD19 PC7, anti-CD14 FITC, anti-CD16 PC5, anti-CD15 PE, anti-CD57 FITC, anti-CD56 PE, anti-CD11c PCP, anti-CD123 PE, anti-CD83 PE, anti-CD38 PE, anti-CD23 ECD and isotype controls (all from Beckman Coulter) for 20 minutes in the dark at +4˚C. Samples were analyzed on the flow cytometer Cytomics FC500 (Beckman Coulter). Data were processed by FlowJo V.10.

## Statistical analysis

Descriptive analysis of collected data and hypothesis testing of observed differences in measured variables were used. Shapiro-Wilk test was employed for the evaluation of normality data distribution. Independent groups t-test and analysis of variance (ANOVA) were used for

comparison between groups. Two-sided *p*-values of less than 0.05 were considered statistically significant. Commercial statistical program SPSS (version 19.0, SPSS Inc., Chicago, IL) was used for data analysis.

## Results

### Baseline characteristics of COVID-19 patients

The median age of patients in MD and SD group was 57 and 62 years, respectively. Among SD cases, there were twice as many men as women (Table 1). Fever (73.3% vs.78.8%), cough (73.3% vs.66.7%) and weakness (36.7% vs. 29.6%) in both patients groups, and shortness of breath (33.3%) in SD group were the most common symptoms upon admission. According to physical examination, 70.4% of SD patients had diminished breath sound compared to 46.7% of MD patients. Crackles were present in 55.6% SD patients. Radiological findings (standard chest X-ray) showed mainly interstitial changes (40.7%), individual pneumonic (44.4%) and diffuse pneumonic foci (55.6%) in SD patients, while no active lesions (30%), interstitial changes (33.3%) and individual pneumonic foci (56.7%) were the most common findings in MD patients. Hematological parameters and biochemical analysis of the whole patient cohort

**Table 1. Clinical characteristics of COVID-19 patients.**

| Characteristic | MD n = 30 | SD n = 27 |
|---|---|---|
| **Age, median (min-max)** | 57(23–88) | 62(45–81) |
| **Gender** | | |
| Male, n (%) | 17(56.7) | 18(66.7) |
| Female, n (%) | 13(43.3) | 9(33.3) |
| **Symptoms** | | |
| Fever, n (%) | 22(73.3) | 21(78.8) |
| Cough, n (%) | 22(73.3) | 18(66.7) |
| Shortness of breath, n (%) | 4(13.3) | 9(33.3) |
| Chest pain, n (%) | 4(13.3) | 3(11.1) |
| Myalgia, n (%) | 2(6.7) | 1(3.7) |
| Headache, n (%) | 2(6.7) | 4(14.8) |
| Weakness, n (%) | 11(36.7) | 8(29.6) |
| Sore throat, n (%) | 4(13.3) | 3(11.1) |
| Loss of smell, n (%) | 1(3.3) | 1(3.7) |
| **Gastrointestinal symptoms, n (%)** | 3(10) | 6(22.2) |
| **Physical examination** | | |
| Normal breath sound, n (%) | 9(30) | 0 |
| Diminished breath sound, n (%) | 14(46.7) | 19(70.4) |
| High–pitched breath sounds, n (%) | 2(6.7) | 1(3.7) |
| Crackles, n (%) | 9(30) | 15(55.6) |
| Wheezing, n (%) | 1(3.3) | 3(11.1) |
| **Chest X-ray** | | |
| Without active lesions, n (%) | 9(30) | 0 |
| Prominent bronchovesicular markings, n (%) | 1(3.3) | 2(7.4) |
| Interstitial changes, n (%) | 10(33.3) | 11(40.7) |
| Individual pneumonic foci, n (%) | 17(56.7) | 12(44.4) |
| Diffuse pneumonic foci, n (%) | 1(3.3) | 15(55.6) |
| Ground-glass opacities, n (%) | 0 | 1(3.7) |
| Homogenous opacities, n (%) | 1(3.3) | 1(3.7) |

**Table 2. Haematological and serum biochemistry parameters in COVID-19 patients with mild and severe disease.**

| Parameters | MD | SD | Normal range | p |
|---|---|---|---|---|
| WBC x10$^6$/mL | 5.4 ± 1.7 | 9.7 ± 4.9 | 3.7–10.0 | <0.0001 |
| Granulocytes x10$^6$/mL (%) | 3.6 ± 1.4 (67.2) | 7.4 ± 5.0 (82.4) | 2.0–7.0 (44–72) | <0.0001 |
| Lymphocytes x10$^6$/mL (%) | 1.1 ± 0.3 (21.1) | 0.6 ±0.4 (9.3) | 0.8–4.0 (20–46) | <0.0001 |
| Monocytes x10$^6$/mL (%) | 0.6 ± 0.2 (10.5) | 0.5 ± 0.4 (6.7) | 0.12–1.2 (2–12) | <0.0001 |
| AST IU/L | 32.3±15.0 | 54.7±31.2 | 0.-40 | 0.017 |
| Albumins g/L | 36.6 ± 4.3 | 29.6 ± 6.4 | 35–52 | <0.0001 |
| LDH U/L | 450.8 ± 99.5 | 919.2 ± 282.4 | 220–450 | <0.0001 |
| CK U/L | 124.7 ± 132.0 | 210.8 ± 144.6 | <190 | 0.035 |
| CRP mg/L | 36.3 ± 41.5 | 119.3 ± 80.9 | <5.0 | <0.0001 |
| PCT ng/mL | 0.1 ± 0.1 | 0.3 ± 0.3 | <5.0 | 0.001 |
| pCO$_2$ kPa | 4.7 ± 0.6 | 6.0 ± 3.1 | 4.7–6 | 0.048 |
| Saturation % | 96.1 ± 2.4 | 89.1 ± 6.7 | 95–97 | <0.0001 |
| K mmol/L | 4.4 ± 0.5 | 3.9 ± 0.6 | 3.5–5.3 | 0.003 |

The numbers represent the means ± standard deviations.

showed that increased granulocytes, glycemia, ALT, LDH, CK, D-dimer, and CRP were the most typical findings, while oxygen saturation and blood pH were under normal values in the majority of patients (S1 Table). The values of laboratory parameters were significantly different among MD and SD cases (Table 2). WBC, granulocyte percent, LDH, CK, CRP, PCT, pCO$_2$ were higher in SD patients, while the lymphocyte and monocyte counts, albumin levels, oxygen saturation, and potassium concentrations were lower among SD patients compared to MD ones.

## Flow cytometry analysis

**Changes in the frequency of peripheral blood cells in COVID-19 patients.** Both mild and severe cases had a higher percent of polymorphonuclear, but lower percentage ratio of mononuclear cells in comparison to healthy controls, and that difference was more profound in severe cases (Table 3).

Although in non-severe patients all parameters were close to, or in the normal range, percent of CD15+ cells (neutrophils) were higher than in controls, and proportions of B lymphocytes (CD19+), monocytes (CD14+) and both helper (CD3+CD4+) and cytotoxic (CD3+CD8 +) T lymphocytes were lower. In patients with severe disease very high neutrophil-to-lymphocyte ratio (17.4) reflected an increase in neutrophil count (90.1%), diminution in B lymphocyte (2.1%) and NK cells count (1.8%), and marked decline of T lymphocyte percentage (1.2%). Very low percent, far below the lower limit of the normal range, was found for both helper (0.8%) and cytotoxic (0.1%) T cells. CD4/CD8 ratio was three times higher in severe patients than in controls (S1 Fig), but without statistical significance. The percentage of monocytes/macrophages was in the normal range in both mild and severe cases, although lower than in control.

**NK cell response differs in MD and SD patients.** The NK cell subpopulation (CD3-CD56+) was assessed for the expression of CD57, a marker of NK cell maturation and activation. The change in NK cell number was statistically significant among the groups ($p = 0.016$). In relation to control, mild cases had a higher number of NK cells (6.3% vs. 4.2% in controls), but about the same percent of CD57+ cells (1.1% in mild cases; 1.2% in controls), while in severe cases both total number of NK cells and percent of activated cells were lower (1.8% and 0.5%, respectively) (Fig 1), even though statistical significance was not reached.

**Table 3. Peripheral Blood Leukocytes (PBL) percentage and absolute count in healthy subjects, patients with mild disease, and patients with severe disease, as shown by flow cytometric analysis.**

| PBL | control | | | | mild | | | | severe | | | | p |
|---|---|---|---|---|---|---|---|---|---|---|---|---|---|
| | median (%) | range (%) | median (x10⁶/mL) | range (x10⁶/mL) | median (%) | range (%) | median (x10⁶/mL) | range (x10⁶/mL) | median (%) | range (%) | median (x10⁶/mL) | range (x10⁶/mL) | |
| Ne/Ly | 1.2 | 1.0–3.2 | 1.9 | 1.4–2.6 | 1.7 | 1.5–2.9 | 3.5 | 1.9–5.2 | 17.4 | 9.0–23.1 | 75.2 | 21.3–155.7 | <0.0001 |
| CD15+ | 49.4 | 37.9–57.4 | 2.6 | 2.0–3.2 | 58.2 | 42.1–64.3 | 2.7 | 1.7–6.2 | 90.1 | 83.1–93.4 | 6.3 | 3.7–18.7 | <0.0001 |
| CD3+ | 24.4 | 4.4–30.0 | 1.1 | 0.3–1.6 | 15.6 | 8.3–32.3 | 0.8 | 0.4–1.5 | 1.2 | 0.6–3.9 | 0.7 | 0–0.4 | <0.0001 |
| CD3+CD4+ | 14.1 | 11.7–19.2 | 0.9 | 0.6–1.1 | 9.1 | 4.9–11.1 | 0.4 | 0.2–0.9 | 0.8 | 0.0–2.4 | 0.1 | 0–0.3 | <0.0001 |
| CD3+CD8+ | 5.8 | 4.9–9.7 | 0.3 | 0.2–0.6 | 2.5 | 0.7–3.1 | 0.1 | 0–0.2 | 0.1 | 0.1–1.4 | 0 | 0–0.1 | <0.0001 |
| CD4+/CD8+ | 2.0 | 1.5–3.9 | 2.0 | 1.5–3.9 | 4.2 | 1.9–13.1 | 4.2 | 1.9–13.3 | 6.5 | 0.1–19.6 | 6 | 0–17 | 0.54 |
| CD19+ | 8.2 | 3.4–12.4 | 0.5 | 0.2–0.7 | 5.2 | 2.2–17.0 | 0.3 | 0.1–1.7 | 2.1 | 1.6–3.7 | 0.2 | 0.1–0.4 | 0.0032 |
| CD3-CD56+ | 4.2 | 4.1–10.3 | 0.3 | 0.2–0.6 | 6.3 | 3.1–8.5 | 0.3 | 0.2–0.6 | 1.8 | 1.1–2.5 | 0.1 | 0.1–0.4 | 0.016 |
| CD14+ | 8.7 | 4.9–11.4 | 0.5 | 0.3–0.8 | 7.7 | 5.8–11.0 | 0.4 | 0.-0.8 | 5.3 | 2.3–11.5 | 0.5 | 0.2–1.1 | 0.042 |

**HLADR expression decreases with the disease severity.** Further analysis demonstrated that the percent of cells expressing HLA-DR was almost two times lower in mild cases than in controls (8.1% vs. 14.9%), and 6.5 times lower in severe cases (2.3%) with statistical significance of $p < 0.0001$. Namely, a decrease of HLA-DR expression, more pronounced in severe cases, was determined in both monocytes (8.2%—controls; 5.7%—mild cases; 2.5%—severe

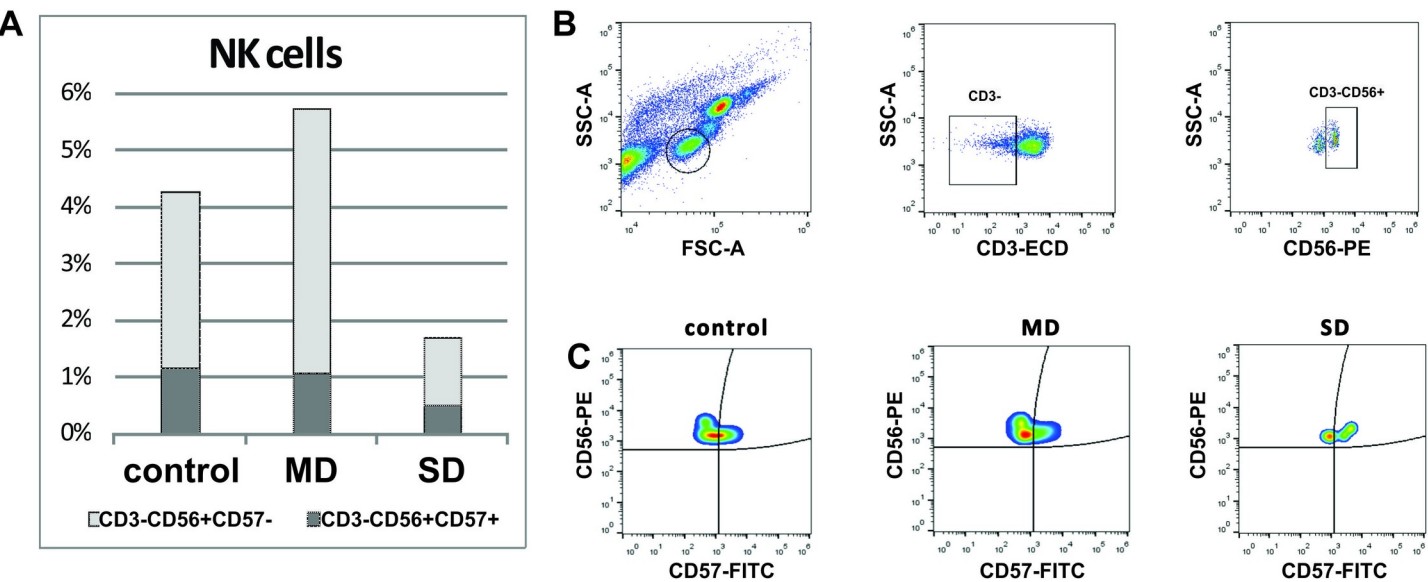

**Fig 1. Differences in NK cell population dependent on severity of the disease.** (A) Bar chart showing the percentage of NK cells in peripheral blood leukocytes and the proportion of CD57⁻ and CD57⁺ cells within NK cell population. (B) Gating strategy: lymphocytes were selected using FS/SS properties; NK cells were selected by the absence of CD3 and CD19. (C) NK cells were defined as activated based on the co-expression of CD57.

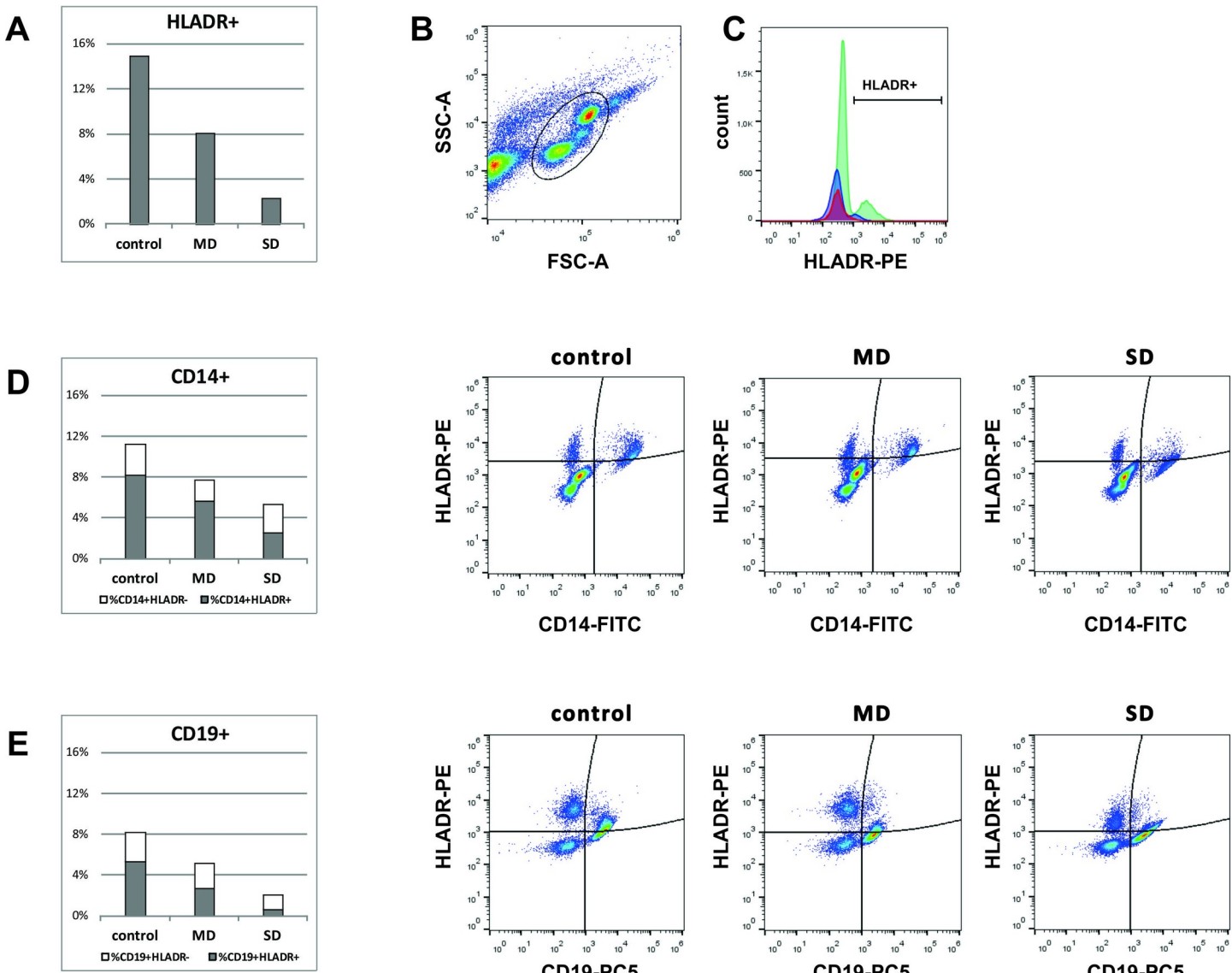

**Fig 2. Decrease of HLA-DR expression is more pronounced in severe cases.** (A) Bar chart showing the percentage of HLA-DR⁺ cells in peripheral blood leukocytes. (B) FSC/SSC dot plot was used to gate leukocytes and to exclude debris. (C) Overlaid histogram presenting HLADR expression in leukocyte population of controls (green), MD (blue) and SD patients (red). (D) Bar chart and representative pseudocolor plots showing percent of HLADR+ and HLADR- cells within monocyte population. (E) Bar chart and representative pseudocolor plots showing percent of HLADR+ and HLADR- cells within B lymphocyte population.

cases, $p<0.0001$) and B lymphocytes (5.3%—controls; 2.6%—mild cases; 0.7%—severe cases, $p = 0.014$) (Fig 2).

**Ups and downs of dendritic cells in COVID-19 patients are correlated with the severity of disease.** The percentage ratio of dendritic cells (Lyn-HLADR+) didn't differ much between controls and mild cases, but in later, there was a lower number of plasmacytoid (CD123+) DCs (0.6% vs. 1.4% in control, $p = 0.0017$), a higher number of myeloid (CD11c+) DCs (1.0% vs. 0.2% in control, $p = 0.0047$), and DCs expressing CD83, an activation marker for antigen-presenting cells (0.7% vs. 0.5% in control) (Fig 3). Contrarily, in severe cases, CD11c+ DCs were almost undetectable and the percent of activated and plasmacytoid DCs was lower than in mild cases and controls.

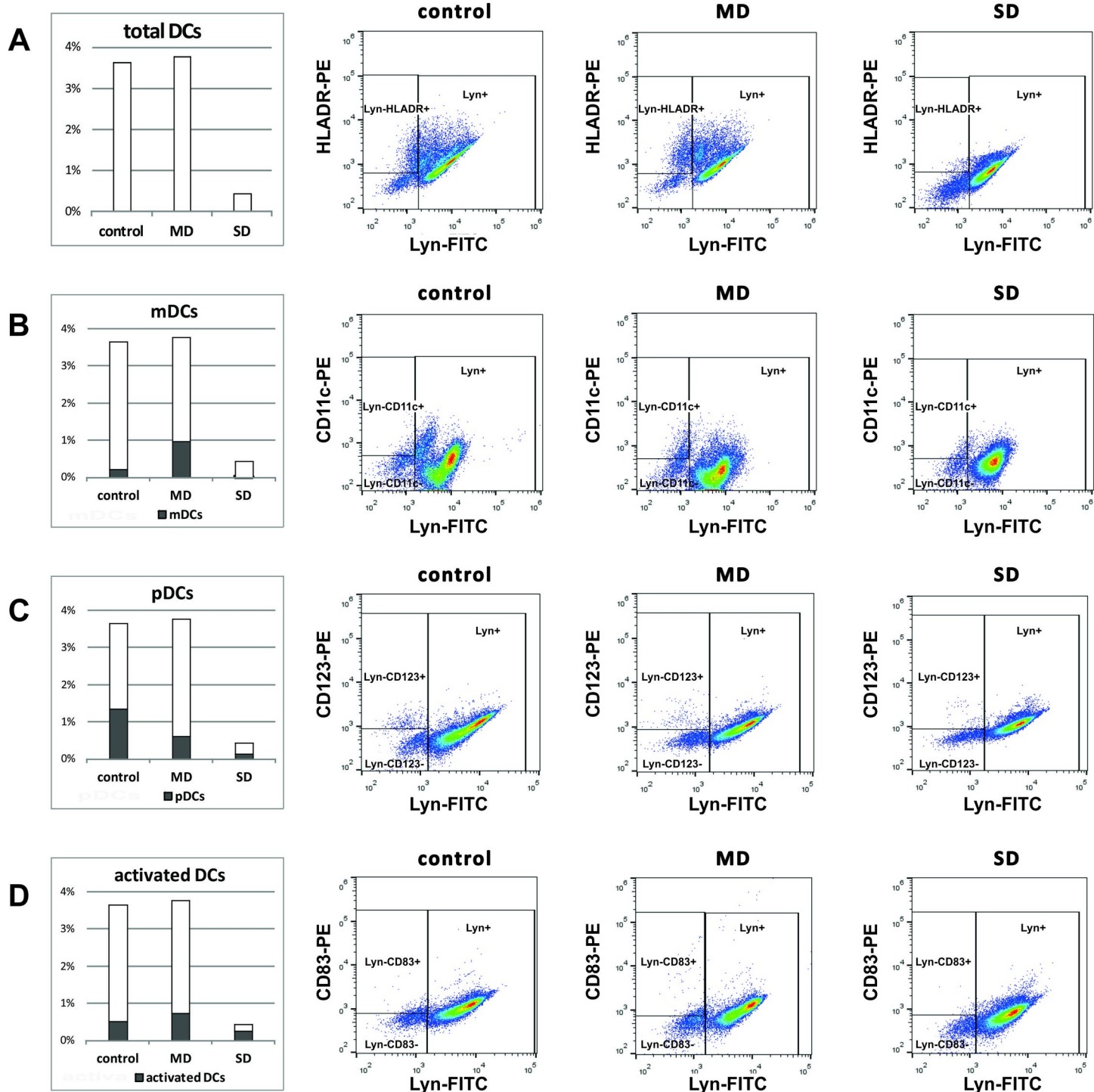

**Fig 3. Dendritic cell subsets differ between the patients with mild (MD) and patients with Severe Disease (SD).** Bar charts (left panel) and representative pseudocolor plots (right panel) showing percent of: (A) overall dendritic cells and (B) myeloid DCs, (C) plasmacytoid DCs and (D) activated DCs within dendritic cell population in controls and patients.

**Perturbations in percentage ratio of monocyte subsets and the expression of mixed M1/ M2 phenotype is more pronounced in SD patients.** Further, by tracking relative expression levels of CD14 and CD16 surface molecules, we examined the proportions of monocyte

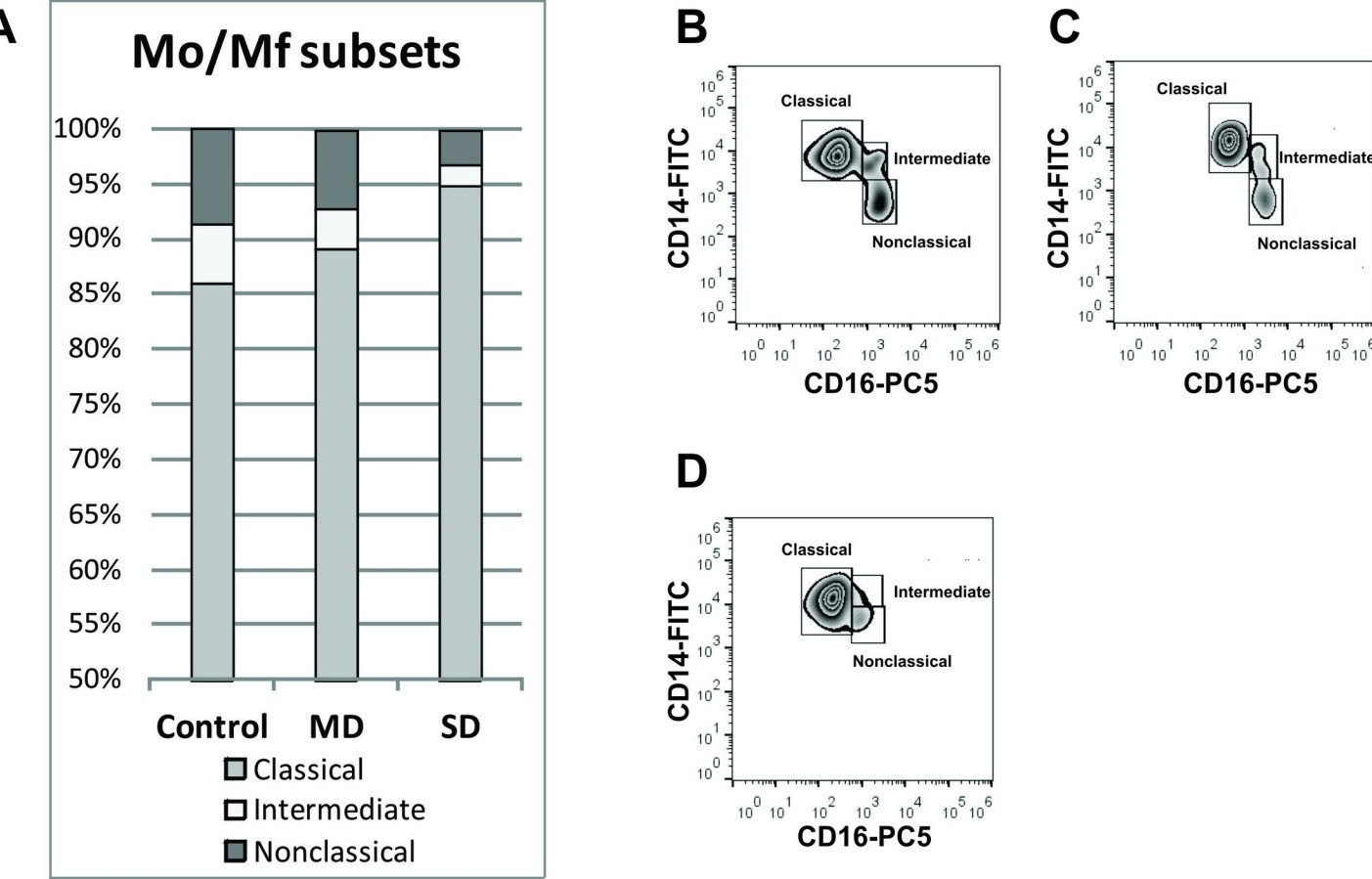

**Fig 4. Percentage ratio of monocyte subsets is altered in COVID-19 patients.** Gating strategy for identifying monocyte subsets is described in S2 Fig. (A) Bar chart presenting proportion of classical, intermediate and nonclassical monocyte subsets in controls and patients. Representative zebra plots are presenting monocyte subsets in (B) healthy controls, (C) patients with mild (MD) and (D) patients with severe disease (SD).

subsets, classical, intermediate, and non-classical (S2 Fig). In relation to control the percent of classical monocytes (CD14$^{high}$CD16-) was higher in mild cases and even higher in severe cases with statistical significance (86.0% in control, 89.2% in mild cases and 94.8% of total monocytes in severe cases, $p = 0.0033$), while the percent of intermediate (CD14$^{high}$CD16+) and non-classical monocytes (CD14$^{low}$CD16+) decreased (Fig 4) (intermediate: 5.5% in control, 3.6% in mild and 1.9% in severe cases, $p = 0.14$; non-classical: 8.6% in control, 7.2% in mild and 3.3% in severe patients, $p = 0.0057$).

Next, the polarization of monocytes was revealed using CD38 as a marker of M1 monocytes, and CD23 typical of M2 monocytes. In the monocyte population of severe cases, we found higher percent of CD38+ (98.3% vs. 94.1% in control and 93.3% in mild cases, $p = 0.039$) and a markedly higher percent of CD23+ cells (10.1% vs. 1.5% in control and 1.6% in mild cases, $p = 0.0032$). Of note, all monocytes positive for CD23, co-expressed CD38 as well (Fig 5).

Cells co-expressing CD23 and CD38 were absent in classical monocyte subset of both controls and patients, but they were present in intermediate and non-classical subsets in patients, especially in severe cases (intermediate: 5.9%—control, 21.5%—mild, 27.7%—severe cases, $p = 0.17$; non-classical: 3.6%—control, 17.1%—mild, 48.5%—severe cases, $p = 0.0021$) (Fig 6).

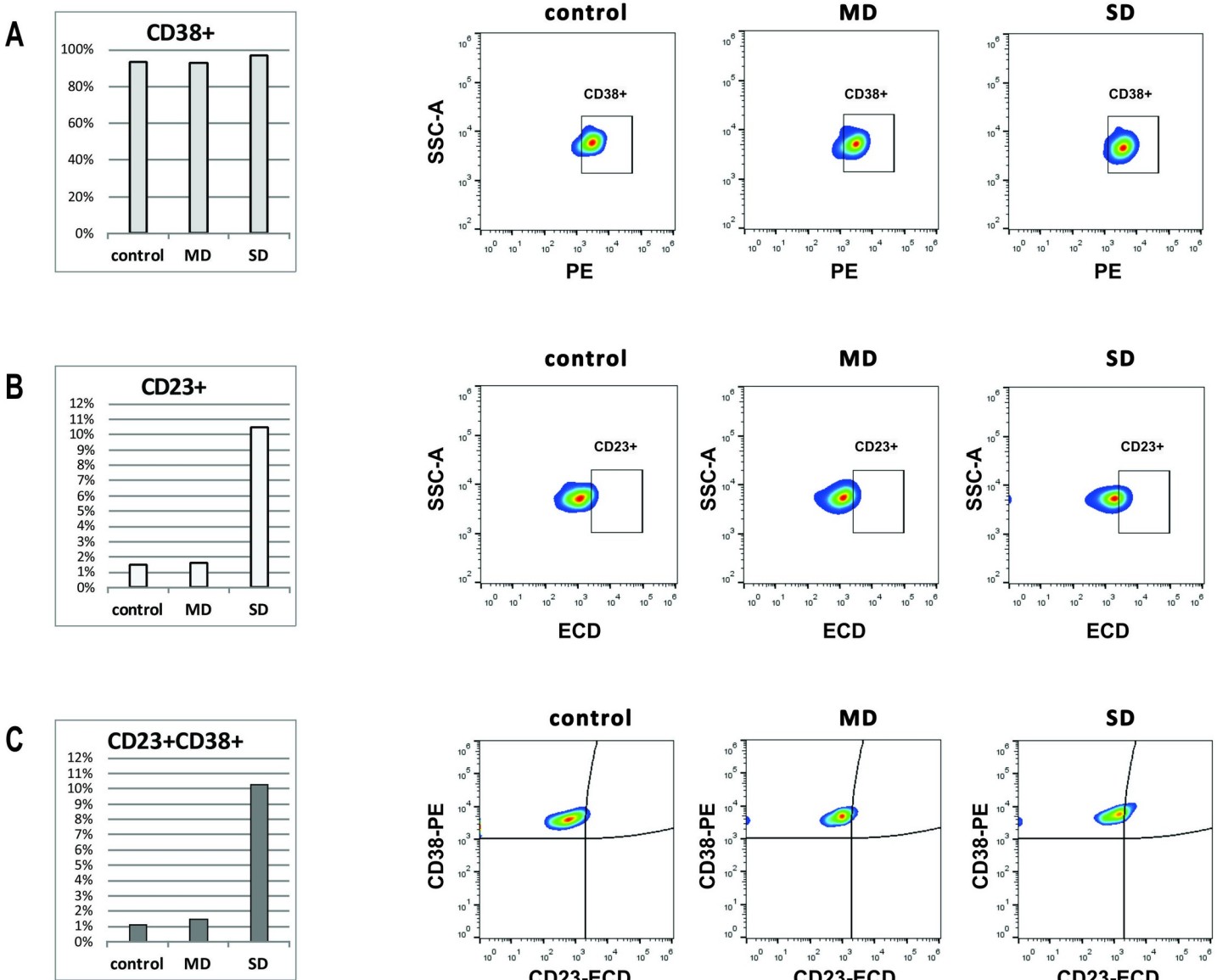

**Fig 5. Co-expression of CD23 and CD38 is the highest in SD patients monocytes.** Monocyte gating strategy is described in S2 Fig. Bar charts (left panel) and representative pseudocolor plots (right panel) are showing percent of (A) CD38+, (B) CD23+ and (C) double positive CD23+CD38+ cells in monocyte population.

## Discussion

COVID-19 is the third emerging coronavirus infectious disease in the 21st century. The virus was introduced from an animal reservoir and met an immunologically naive human population. A number of studies have described changes within innate and adaptive immune response in SARS-Cov-2-infected patients, but there are still many unknowns. In our study, we recorded baseline characteristics of COVID-19 patients and analyzed changes in peripheral blood cell populations in relation to control healthy subjects.

SARS-Cov-2 infection provokes sustained cytokine and chemokine secretion, leading to severe lung injury, multiorgan failure, immune dysfunction, and mortality. Our results are in line with previous findings showing that these alterations came under the signs of a common respiratory infection such as fever, cough, and fatigue [14] and that most patients had

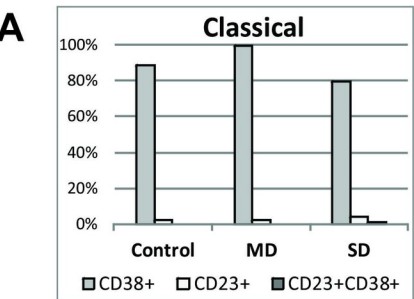 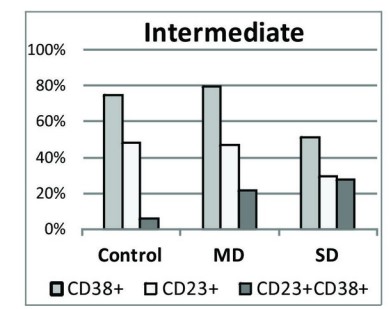 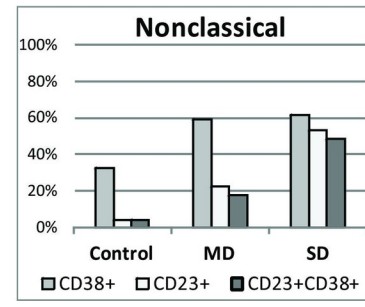

**Fig 6. Mixed M1/M2 phenotype prevails in intermediate and nonclassical monocyte subsets of SD patients.** Bar charts are showing percent of CD38+, CD23+ and double positive CD23+CD38+ cells in (A) classical, (B) intermediate and (C) nonclassical subsets of monocytes in controls, MD and SD patients.

granulocytosis, elevated infection-related and organ-injuries-related biomarkers, including LDH, CK, ALT, which were higher among severe cases. Also, we found elevated levels of D-dimer and CRP, parameters that have been reported to be associated with the severity of disease [4]. The cytokine storm is thought to be responsible for multiorgan damage and elevated organ-injuries-related biomarkers. However, correlation analysis between lymphocyte subsets and biochemical markers showed that most biochemical markers indicating organ damage are negatively correlated with lymphocyte counts in SARS-Cov-2 patients, which is not the case in patients with pneumonia of other etiologies. This finding highlights that the potential cause of multi-organ injury is the virus itself [15].

The common findings of previous research were high percent of neutrophils, lymphopenia, and high neutrophil-to-lymphocyte ratio in COVID-19 patients in comparison to healthy subjects, whereas this difference was more radical in severe disease cases, which is in consent with our results [6, 16–19]. Lymphopenia is one of the most salient markers of COVID-19 and it seems to arise as the result of the reduction of all lymphocyte populations, including CD4+ and CD8+ T cells, B cells, and NK cells. In line with our results, Zhou et al. [18] have described that the decline in CD4+ T lymphocyte count is significant in both severe and mild patients, while the decrease of CD+ 8 cells was more profound in severe patients. Still, there is a report that the reduction in CD4+ T cells is much greater in severe cases [6]. As previously described, we found a greater decrease in CD8+ than in CD4+ subpopulation, and the decrease in B cell percentage, that was more expressed in severe cases. It's noteworthy that the decline in the frequency of all lymphocyte populations is more profound in COVID-19 patients in comparison with non-SARS-Cov-2-pneumonia patients [15]. One of the possible causes of lymphopenia is the sequestration of lymphocytes in the lung tissue, at the site of infection. The autopsy showed that infiltrating cells were mostly monocytes and macrophages, with multinucleated giant cells, and a few lymphocytes, being mainly CD4+ T cells [20]. Further examination also revealed that the number of trilineage in the bone marrow and lymphocytes in the spleen and lymph nodes are all significantly reduced. These facts indicated that lymphopenia cannot be attributed only to the tissue redistribution of lymphocytes and brought to the foreground the possible direct effect of the virus on immune cells. MERS-CoV is known to directly infect human T lymphocytes and activate the extrinsic or intrinsic apoptotic pathway, but does not replicate. Although ACE2 is not expressed on lymphocytes, as the main site of SARS-Cov-2 binding, recent research has reported a novel invading route of SARS-Cov-2. Namely, it has been noted that SARS-Cov-2 can infect lymphocytes through spike protein (SP) interaction with lymphocyte's CD147 protein, highly expressed on activated T and B cells, but also on dendritic cells, monocytes and macrophages [21, 22]. Whether virus induces direct cytopathic effects is not yet elucidated.

NK cells, as a member of innate immunity, provide crucial early defense against viral infections. Contrarily to other reports [7, 8, 16], our results showed that in mild COVID-19 patients the percentage of total NK cells was higher in comparison to control, and the percent of activated cells was preserved, but in severe cases, both values were remarkably decreased. It has been described that type I IFN is required for NK cell activation [23]. Low percent of dendritic cells in severe patients, as described, may contribute to decreased secretion of type 1 IFN, resulting in a decline of NK cell activation. Further, the overproduction of IL-6 plays a role in the reduced activity of NK cells in mimicked viral infection *in vivo* [24]. Recent findings state that IL-6 level is high in severe COVID-19 patients, remaining low in mild cases [5], and negatively correlates with NK cell count and activity [7]. Moreover, some authors noted upregulated expression of the inhibitory receptor NKG2A on NK cells in the early stage of COVID-19 [8].

Abnormalities in cells that bridge innate and adaptive immunity, and regulate the latter may be responsible for the reduction of lymphocyte number and function. Key roles in immune response regulation and antigen presentation play dendritic cells (DCs). A number of ways in which viruses affect the adequate response of DCs have been described [25]. In our study, we found that the percentage of overall DCs, as well as the percent of myeloid and activated DCs, was higher in mild cases in relation to control, indicating their preserved antigen-presenting function. Also, Cao et al. [26] have demonstrated that in Influenza virus-induced differentiation of monocytes into mDCs, those cells, unlike classic mDCs, secrete chemoattractants for monocytes and type I IFN. Contrarily, in severe cases, the percent of total DCs and analyzed subpopulations was lower. Functional activation analysis of DCs in SARS-CoV infection were inconclusive showing both activation [9], and the absence of activation [27]. The consequence of low degree dendritic cell activation is the insufficiency of costimulatory molecules, necessary for survival during TCR engagement, which partly explains the reduction of T lymphocytes dying by apoptosis in the absence of adequate signaling. Reduced percentage of plasmacytoid dendritic cells that we found in both mild and severe patients, suggests that the adequate response to viral infection was profoundly disrupted, considering that plasmacytoid DCs represent the main source of type I IFN. A similar effect was observed in SARS-CoV infection, where DCs failed to trigger a strong type I IFN response, implicating that the virus circumvents the activation of the innate immune system [28]. SARS-CoV also promoted a moderate increase in the production of IL-6 in DCs [29].

The expression of HLA-DR molecules is restricted to the cells with a specialized role in antigen presentation. Therefore, the extent of HLA-DR expression on monocytes and B cells indicates their ability for antigen-presentation. In our study overall expression of HLA-DR molecule on peripheral blood mononuclear cells was extremely downregulated among both mild and severe COVID-19 patients. The decline in monocyte HLA-DR expression in SARS-CoV-2 infection has been described in recent studies [11, 19], pursuant to our results. It should be underlined that we also found reduced HLA-DR expression on B lymphocytes. Of note, in our cohort HLA-DR expression was even 6.5 times lower in severe patients. Suppression of HLA-DR expression on monocytes below the threshold value (<30% HLA-DR + monocytes) has been accepted as a definition of immunoparalysis that occurs in lethal conditions such as sepsis and represents a predisposition for superinfection with a variety of pathogens [30]. Giamarellos-Bourboulis et al. [19] proposed that one of the drivers of decreases in HLA-DR expression is IL-6 concentration, based on finding that IL-6 concentration is reciprocal to HLA-DR expression.

The percentage ratio of monocytes in COVID-19 patients was in the normal range, but flow cytometric analysis showed that they are different from those in healthy subjects. Although we didn't find FSC-high and SSC-high monocyte population described by some

authors [10, 11], we revealed that the proportion between certain subsets of monocytes was disturbed in patients in comparison to controls. In severe cases we found decline in intermediate (CD14$^{high}$CD16+) and non-classical (CD14$^{low}$CD16+) monocytes, along with robust induction of classical (CD14$^{high}$CD16-) monocytes. Similar redistribution of monocyte subsets was also present in mild cases, but in a more tenous manner. In the peripheral blood of healthy humans, classical monocytes are the major population of monocytes (80–95%) [31]. The main function of these so-called „inflammatory" monocytes is phagocytosis and secretion of proinflammatory cytokines. Besides, they are the primary source of monocyte-derived DCs and tissue macrophages. Intermediate monocytes (2–8%), which also have inflammatory properties, are the main ROS producers and have the highest expression of MHC II class molecules (HLA-DR), acting as specialized antigen-presenting cells. Non-classical monocytes (2–11%) are "patrolling" monocytes that travel across blood vessels to scavenge dead cells and pathogens. In contact with infectious agents, they produce inflammatory cytokines and chemokines that recruit neutrophils, and subsequently clear resulting debris and promote healing and tissue repair. During inflammation, non-classical monocytes can extravasate to inflamed tissue and differentiate to inflammatory macrophages. The three monocyte subsets have distinct roles in response to different stimuli, i.e., during homeostasis, inflammation, and tissue repair. Although there are still not many results on this topic, and the existing ones are, to a certaine extent, contradictory, our findings are in line with many published results [32–34]. Prominet decrease of intermediate and non-classical monocytes in blood, esspecialy in severe patients, might be a result of the monocyte migration into the lungs, as described by Sanchez-Cerrillo et al [34]. This hypothesis is supported by the fact that 70% of SD patients and 10% of MD patients assigned to flow cytometry had diffuse pneumonic foci. In contrast to our reports, Zhang and Zhou [10, 18] found an increase in CD16+ monocytes in COVID-19 patients. Divergence of immune response might be a repercussion of viral mutations. Namely, several reports result showed that, due to SARS-CoV-2 rapidly spreading across countries, new mutation hotspots are emerging in different parts of the genome, indicating a presence of different viral strains in Europe and Asia, with European strains being more virulent [35–38]. Also, sex differences in monocyte subsets redistribution are described [39].

CD38 is multifunctional protein expressed on the cells of innate, as well as adoptive immunity. Constitutive expression of CD38 on peripheral blood monocytes is upregulated in inflammatory conditions and increase in surface density of this molecule contributes to the proinflammatory phenotype of monocytes [40]. Expression of CD23, a low affinity IgE receptor, in human monocytes is upregulated by IL-4 and reflects anti-inflammatory phenotype [41]. Therefore, to further define functional changes in monocyte subsets during infection, we observed the expression of surface molecules CD38 and CD23. Our results showed a high expression of both CD38 and CD23 on monocytes in patients with severe disease. While in controls double positive cells were absent or present in very low percent, in COVID-19 patients a large number of monocytes co-expressing CD23 and CD38 were found in intermediate and nonclassical subsets. Importantly, in SD patients in both subsets almost all CD23+ cells were positive for CD38 as well. A concept of M1 and M2, pro- and anti-inflammatory monocytes, mirroring M1/M2 macrophage polarization, was introduced by Fukui et al. [42]. Ever since, this concept was accepted by other researchers [43–45]. Mixed M1/M2 phenotype has been described in chronic infections, autoimmune diseases, cancer, and disorders associated with fibrosis [46]. It is postulated that these monocytes have both proinflammatory and tissue repair functions. A redistribution of monocyte subsets, with M1/M2 non-classical monocytes preferentially migrating to lungs [34] could promote both inflammation and fibrosis as a tissue repair mechanism. In line with that, pulmonary fibrosis as a complication of severe COVID-19 can also be a significant cause of mortality in COVID-19 patients [47]. Considering the central role

that monocytes play in the pathogenesis of cytokines storm in lung damage of COVID-19, here we present both conformation of some already available evidence and add some novelty about substantial phenotypical alterations of monocytes in COVID-19 patients.

Overall, pursuant to the previous findings and based on our results, we can speculate that SARS-CoV-2 virus causes dismantling of the immune response, but the resulting alterations differ in mild and in severe patients. In patients who didn't develop severe symptoms, a decline in lymphocyte number is lesser, and the innate immunity is preserved. Despite a decrease in the percent of HLA-DR-expressing B cells and monocytes, the number of total DCs, mDCs (possibly type I IFN producing cells), and activated DCs is higher than in control, pointing to their sustained functions. NK cells, that play a key role in host defense against viral infections, are also present in higher number, and activated cells are immanent as in healthy subjects. In monocyte population expression of M2 marker CD23 is low, as in healthy controls. On the other hand, in severe cases, both arms of immune defense, innate and adoptive, are affected. The number of T and B lymphocytes is dramatically decreased, as well as the number of NK cells and DCs (both total and activated). The number of cells expressing HLA-DR is drastically lesser in severe cases, depicting the inability of antigen-presenting cells to activate T lymphocyte response. The virus affects the monocyte population as well. The percentage ratio of intermediate and nonclassical monocytes is reduced, pointing to impaired functional maturation. An increase in the percent of cells coexpressing markers of M1 and M2 monocytes points to prolongation of inflammation and evolution of fibrosis as a repair mechanism, that damages lung parenchyma and potentially increases the risk of worse clinical outcomes. Altogether, our results depict the devastation of host defense in severe patients and altered, but a more efficient immune response in patients with mild/moderate symptoms.

## Supporting information

**S1 Table. Haematological and serum biochemistry parameters in COVID-19 patients.** The numbers represent the means ± standard deviations.
(DOCX)

**S1 Fig. Flow cytometry analysis of T cells in peripheral blood leukocytes of healthy controls, patients with mild (MD) and patients with Severe Disease (SD).** (A) Gating strategy: lymphocytes were selected using FS/SS properties; T lymphocytes were identified based on expression of CD3. (B) CD3 vs. CD8 pseudocolor plots showing CD3+CD8+ cells and (C) CD3 vs. CD4 pseudocolor plots showing CD3+CD4+ cells in T lymphocyte population of controls and patients.
(TIF)

**S2 Fig. Monocyte gating strategy.** (A) FS *vs*. SS plot: Wide selection of monocytes depending on FS/SS properties. (B) Pseudocolor CD16 *vs*. CD14 plot: Gating to select monocytes depending on characteristic "inverted L" shape. (C) Pseudocolor CD16 *vs*. HLA-DR plot: Gating to select HLA-DR$^+$ cells and to remove NK cells. (D) Pseudocolor CD14 *vs*. HLA-DR plot: Gating to exclude B cells (HLA-DR$^{high}$/CD14$^{low}$). (E) Pseudocolor CD16 *vs*. CD14 plot: Gating to select classical (CD14$^{high}$CD16$^-$), intermediates (CD14$^{high}$CD16$^+$) and non-classical (CD14$^{low}$CD16$^+$) monocytes. (F) Zebra CD16 *vs*. CD14 plot: Selected monocytes redisplayed on CD16 vs. CD14 zebra plot to visualize monocyte subsets.
(TIF)

**S1 Dataset.**
(XLS)

## Acknowledgments

We thank the health workers of the Clinical Centre Kragujevac, doctors and especially technicians, who selflessly, in difficult moments of the struggle for the lives of patients, supported our study.

## Author Contributions

**Conceptualization:** Sanja Matic, Dejan Baskic.

**Formal analysis:** Sanja Matic, Suzana Popovic, Natasa Djordjevic, Dragan Milovanovic, Dejan Baskic.

**Investigation:** Sanja Matic, Predrag Djurdjevic, Danijela Todorovic, Dejana Ruzic Zecevic, Marina Petrovic, Maja Sazdanovic.

**Project administration:** Dejan Baskic.

**Resources:** Zeljko Mijailovic, Predrag Sazdanovic, Nenad Zornic, Vladimir Vukicevic, Ivana Petrovic, Snezana Matic, Marina Karic Vukicevic.

**Supervision:** Dejan Baskic.

**Writing – original draft:** Sanja Matic, Suzana Popovic.

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
