## [Decision Letter · Decision Letter 0]

10 Nov 2020

PONE-D-20-31462

SARS-CoV-2 infection induces mixed M1/M2 phenotype in circulating monocytes and alterations in both dendritic cell and monocyte subsets

PLOS ONE

Dear Dr. Popovic,

Thank you for submitting your manuscript to PLOS ONE. After careful consideration, we feel that it has merit but does not fully meet PLOS ONE’s publication criteria as it currently stands. Therefore, we invite you to submit a revised version of the manuscript that addresses the points raised during the review process.

First of all, I must apologize for the delay in getting a response back to you that was mediated by a failure to receive a response from one of the reviewers. As you may read in the critiques provided by the 2 reviewers, both of them have some concerns hat appear to a large extent technical in nature. Please pay close attention to the suggested changes recommended by the 2 reviewers and address each of these in your revised submission.

We look forward to receiving your revised manuscript.

Kind regards,

Aftab A. Ansari, PhD

Academic Editor

PLOS ONE

Journal Requirements:

2. Please amend your list of authors on the manuscript to ensure that each author is linked to an affiliation. Authors’ affiliations should reflect the institution where the work was done (if authors moved subsequently, you can also list the new affiliation stating “current affiliation:….” as necessary).

Reviewers' comments:

Reviewer's Responses to Questions

**Comments to the Author**

1. Is the manuscript technically sound, and do the data support the conclusions?

Reviewer #1: Partly

Reviewer #2: No

2. Has the statistical analysis been performed appropriately and rigorously? 

Reviewer #1: Yes

Reviewer #2: Yes

3. Have the authors made all data underlying the findings in their manuscript fully available?

Reviewer #1: Yes

Reviewer #2: No

4. Is the manuscript presented in an intelligible fashion and written in standard English?

Reviewer #1: Yes

Reviewer #2: Yes

5. Review Comments to the Author

Reviewer #1: Sanja et al. present an observational study which characterizes changes in circulating immune cells in COVID-19 patients based on severity of disease. This adds to a growing body of literature in this area. The article is well-written, and the methodology appears rigorous where it is sufficiently described. I had several concerns, as follows (in approximately the order they appear in the manuscript):

(1) References to preprints should be updated where they have been subsequently published. For example, references 5, 10, 12, etc.

(2) Please specify anticoagulant(s) used in blood collection.

(3) How were subjects selected for flow cytometry analysis, given that this is only a subset of the total patient sample.

(4) Was blood collection standardized by time point or other means (e.g., days from hospitalization or symptom onset)? Observational studies such as these have a limitation in that they may be measuring a kinetic response rather than a severity response, if this is not controlled.

(5) It would be useful to have Table 1 information split by severity. For example, were patients with more severe disease older?

(6) Please correct D-dimmer to D-dimer. There are several instances of this.

(7) Monocyte phenotyping in COVID-19 has yielded vastly different results for different groups, with some showing increased CD16+ subsets, and some (as here) showing decreases. Trafficking to the lungs is one possible explanation, but it would be worthwhile to explore the differences between this study and studies showing divergent results in the discussion.

(8) CD23 and CD38 results in monocytes require much more thorough citation and expanded discussion. M1 and M2 phenotypes are not generally used as descriptors for monocytes (rather for macrophages), and these markers are not well-characterized in circulating monocytes. Additionally, since all CD23+ monocytes also expressed CD38, this suggests these markers are not necessarily reflecting the phenotype suggested in the paper.

(9) Please clearly label axes in the figures. Some are hard to interpret.

Reviewer #2: General Comments:

In this manuscript, Sanja et al. described the immunological events in the blood associated with severity of SARS-CoV-2 infection particularly with a focus on innate immunity responses. They indicated that severity of the disease correlated with the decrease in numbers of T and B lymphocytes, DCs, NK cells and HLA-DR expressing cells. Moreover, they determined that the monocyte population was disturbed for expression of M1 and M2 monocyte markers in the intermediate and non-classical subsets. Finally, they concluded that both innate and adoptive immune responses are heavily affected in patients with severe disease but that inmate immunity is preserved in patients with milder symptoms per only slight decreases in the lymphocyte population. A concern, unfortunately, was that the data presentation and flow cytometry analyses should be improved to satisfactorily support the conclusions drawn from this study.

Specific Comments:

1. The figure legends should be described and placed outside of the result section.

2. The results section should be described using specific subtitles that indicate the main massage for each section of the study presented.

3. The control data described in S1 Table should be included in Table 2 of the main text. In addition to the % of granulocytes, lymphocytes and monocytes, the absolute numbers of cells/ml would contribute to more informative results.

4. As also suggested for Table 2, the addition of the absolute numbers of cells/ml would be more informative for data presented in Table 3.

5. The entire data presentation and analyses described in Figure 1 should be re-done. None of the figures from panels A-E are informative. For each modified figure, please indicate the take-home message or conclusion for each. To better represent the phenotype changes of NK cells related to severity of disease, the percentage of CD57+ NK cells should be used instead of % of NK cell subsets among leukocytes (F).

6. Figures 2 and 3 should also be modified as recommended for Figure 1.

7. The overlayed figures in Figure 5 and 6 (A-C) are not informative and very difficult to interpret. To better represent the phenotype change of different subsets of monocytes with severity of disease, the percentages of CD38+, CD23+ and CD23/38+ cells should be used for each cell subset.

8. Supplement Figures 1 and 2 should also be modified to clearly indicate the gating strategy of each of the different cell subsets.

6. PLOS authors have the option to publish the peer review history of their article (what does this mean?). If published, this will include your full peer review and any attached files.

Reviewer #1: No

Reviewer #2: No

---

## [Author Response · Author response to Decision Letter 0]

15 Dec 2020

Reviewer #1: 

Sanja et al. present an observational study which characterizes changes in circulating immune cells in COVID-19 patients based on severity of disease. This adds to a growing body of literature in this area. The article is well-written, and the methodology appears rigorous where it is sufficiently described. I had several concerns, as follows (in approximately the order they appear in the manuscript):

1. References to preprints should be updated where they have been subsequently published. For example, references 5, 10, 12, etc.

Re: Among reprint references, references 10 is corrected due to recent publishing. Preprint references 5, 11, 12, 14, 15, 18 and 21 are not published yet on 8th December 2020 at 20:00h.

2. Please specify anticoagulant(s) used in blood collection.

Re: Anticoagualnt used in blood collection is specified in Patients and method section Flow cytometry (EDTA).

3. How were subjects selected for flow cytometry analysis, given that this is only a subset of the total patient sample.

Re: Patients were selected for flow cytometry analysis by creating a randomized list in Microsoft Excel for both groups of patients (SD and MD). The first ten patients from both groups were included in the analysis. The sample size was based on a study of a similar design (Gupta R, Gant VA, Williams B, Enver T. Increased complement receptor-3 levels in monocytes and granulocytes distinguish COVID-19 patients with pneumonia from those with mild symptoms. International Journal of Infectious Diseases. 2020;99:381-5.).

4. Was blood collection standardized by time point or other means (e.g., days from hospitalization or symptom onset)? Observational studies such as these have a limitation in that they may be measuring a kinetic response rather than a severity response, if this is not controlled.

Re: We have stated in manuscript, section Patients and methods, subsection Data sampling that blood samples were collected at admission. To be more precisely, we preformulated in present manuscript that blood speciments were collected on the first day of hospitalisation. Thank you for the suggestion.

5. It would be useful to have Table 1 information split by severity. For example, were patients with more severe disease older?

Re: Table 1 informations are splited by severity. According to changed concept of Table 1, text of manuscript in section Results, subsection Baseline Characteristics of COVID-19 Patients was adapted.

6. Please correct D-dimmer to D-dimer. There are several instances of this.

Re: D-dimmer is corrected to D-dimer throughout the manuscript.

7. Monocyte phenotyping in COVID-19 has yielded vastly different results for different groups, with some showing increased CD16+ subsets, and some (as here) showing decreases. Trafficking to the lungs is one possible explanation, but it would be worthwhile to explore the differences between this study and studies showing divergent results in the discussion.

Re: The differences between this study and studies showing divergent results are expored and interpreted in discussion on pages 19 and 20.

8. CD23 and CD38 results in monocytes require much more thorough citation and expanded discussion. M1 and M2 phenotypes are not generally used as descriptors for monocytes (rather for macrophages), and these markers are not well-characterized in circulating monocytes. Additionally, since all CD23+ monocytes also expressed CD38, this suggests these markers are not necessarily reflecting the phenotype suggested in the paper.

Re: Short description of CD38 and CD23 molecules and the supporting references are now included in section Discussion (next to the last paragraph). 

 Regarding M1 and M2 monocytes, in the same paragraph we included a statement: “A concept of M1 and M2, pro- and anti-inflammatory monocytes, mirroring M1/M2 macrophage polarization, was introduced by Fukui et al. Ever since, this concept was accepted by other researchers.” with supporting references. 

 In order to introduce more clearly the presence of mixed populations of monocytes Figure 6 is revised in form of more informative bar charts that present the percentages of cells expressing CD38+, CD23+ and CD23+CD38+ in each cell subset in controls, MD and SD patients. Our conclusion about the presence of monocytes with mixed M1/M2 phenotype in COVID-19 patients now can be more clearly observed. 

9. Please clearly label axes in the figures. Some are hard to interpret.

Re: In revised Figures all axes are clearly labeled.

Reviewer #2: 

General Comments:

In this manuscript, Sanja et al. described the immunological events in the blood associated with severity of SARS-CoV-2 infection particularly with a focus on innate immunity responses. They indicated that severity of the disease correlated with the decrease in numbers of T and B lymphocytes, DCs, NK cells and HLA-DR expressing cells. Moreover, they determined that the monocyte population was disturbed for expression of M1 and M2 monocyte markers in the intermediate and non-classical subsets. Finally, they concluded that both innate and adoptive immune responses are heavily affected in patients with severe disease but that inmate immunity is preserved in patients with milder symptoms per only slight decreases in the lymphocyte population. A concern, unfortunately, was that the data presentation and flow cytometry analyses should be improved to satisfactorily support the conclusions drawn from this study.

Specific Comments:

1. The figure legends should be described and placed outside of the result section.

Re: Figure legends are removed from the main text and placed at the end of the manuscript.

2. The results section should be described using specific subtitles that indicate the main massage for each section of the study presented.

Re: Section Results is divided into subsections by subtitles that describe the main findings of presented set of data. 

3. The control data described in S1 Table should be included in Table 2 of the main text. In addition to the % of granulocytes, lymphocytes and monocytes, the absolute numbers of cells/ml would contribute to more informative results.

Re: Control data are included in Table 2. Absolute numbers (cells/ml) of granulocytes, lymphocytes and monocytes are included in Table 2 and S1 Table.

4. As also suggested for Table 2, the addition of the absolute numbers of cells/ml would be more informative for data presented in Table 3.

Re: Absolute numbers (cells/ml) of peripheral blood leukocytes are included in Table 3. Due to the large format of the revised Table 3, layout of page 11 in manuscript is changed from portrait to landscape.

5.The entire data presentation and analyses described in Figure 1 should be re-done. None of the figures from panels A-E are informative. For each modified figure, please indicate the take-home message or conclusion for each. To better represent the phenotype changes of NK cells related to severity of disease, the percentage of CD57+ NK cells should be used instead of % of NK cell subsets among leukocytes (F).

Re: Take-home message is now indicated in Figure’s captions. Figure 1 is revised. Overlaid histograms and contour plots are replaced with representative pseudocolor plots presenting gating strategy and analysis of overall and activated NK cells in controls, MD and SD patients separately. To better present the phenotype changes of NK cells related to severity of disease, on bar chart CD57+ and CD57- cells are shown inside the bars presenting percent of overall NK cell population, therefore percent of these cells can be observed both within leukocyte and NK cell populations. 

6. Figures 2 and 3 should also be modified as recommended for Figure 1.

Re: Figures 2 and 3 have been revised following the example of Figure 1. Overlaid histograms and contour plots are replaced with representative pseudocolor plots presenting gating strategy and analysis of overall HLADR+ population in Figure 2, as well as overall DCs, pDCs, mDCs and activated DCs in Figure 3, in controls, MD and SD patients separately. On bar charts in Figure 2 percent of HLADR+ and HLADR- populations are shown inside the bars presenting percent of overall CD14+ and CD19+ population. In Figure 3, separate pDCs, mDCs and activated DCs are presented in perspestive of overall DCs on bar charts. 

7. The overlayed figures in Figure 5 and 6 (A-C) are not informative and very difficult to interpret. To better represent the phenotype change of different subsets of monocytes with severity of disease, the percentages of CD38+, CD23+ and CD23/38+ cells should be used for each cell subset. 

Re: In Figure 5 overlaid contour plots are replaced with representative pseudocolor plots separately for controls, MD and SD patients. The percentages of CD38+, CD23+ and CD23+CD38+ cells in each cell subset are presented by bar charts in Figure 6.

8. Supplement Figures 1 and 2 should also be modified to clearly indicate the gating strategy of each of the different cell subsets.

Re: In Suppl. Figure 1 gating strategy (A) and CD4+ and CD8+ T lymphocytes (B and C) are presented by representative pseudocolor plots separately for controls, MD and SD patients. Gating strategy for monocytes and monocyte subsets is presented step-by-step in Suppl. Figure 2 and described in figure legend.

---

## [Editor Report · Decision Letter 1]

17 Dec 2020

SARS-CoV-2 infection induces mixed M1/M2 phenotype in circulating monocytes and alterations in both dendritic cell and monocyte subsets

PONE-D-20-31462R1

Dear Dr. Popovic,

We’re pleased to inform you that your manuscript has been judged scientifically suitable for publication and will be formally accepted for publication once it meets all outstanding technical requirements.

Kind regards,

Aftab A. Ansari, PhD

Academic Editor

PLOS ONE
---

## [Editor Report · Acceptance letter]

21 Dec 2020

PONE-D-20-31462R1 

SARS-CoV-2 infection induces mixed M1/M2 phenotype in circulating monocytes and alterations in both dendritic cell and monocyte subsets 

Dear Dr. Popovic:

I'm pleased to inform you that your manuscript has been deemed suitable for publication in PLOS ONE. Congratulations! Your manuscript is now with our production department. 

Kind regards, 

on behalf of

Dr. Aftab A. Ansari 

Academic Editor

PLOS ONE